# Small Extracellular Vesicles Derived from Lipopolysaccharide-Treated Stem Cells from the Apical Papilla Modulate Macrophage Phenotypes and Inflammatory Interactions in Pulpal and Periodontal Tissues

**DOI:** 10.3390/ijms26010297

**Published:** 2024-12-31

**Authors:** Solène Tessier, Boris Halgand, Davy Aubeux, Joëlle Véziers, Angélique Galvani, Juliette Jamoneau, Fabienne Pérez, Valérie Geoffroy, Alexis Gaudin

**Affiliations:** 1Nantes Université, Oniris, Univ Angers, Inserm, Regenerative Medicine and Skeleton, RMeS, UMR 1229, F-44000 Nantes, France; solene.tessier@univ-nantes.fr (S.T.); davy.aubeux@gmail.com (D.A.); angelique.galvani@univ-nantes.fr (A.G.); juliette.jamoneau@etu.univ-nantes.fr (J.J.); valerie.geoffroy@univ-nantes.fr (V.G.); 2Nantes Université, Oniris, CHU Nantes, Inserm, Regenerative Medicine and Skeleton, RMeS, UMR 1229, F-44000 Nantes, France; boris.halgand@univ-nantes.fr (B.H.); joelle.veziers@univ-nantes.fr (J.V.); fabienne.perez@univ-nantes.fr (F.P.)

**Keywords:** extracellular vesicles, dental mesenchymal stem cells, immunomodulation, inflammation, macrophage

## Abstract

Inflammation significantly influences cellular communication in the oral environment, impacting tissue repair and regeneration. This study explores the role of small extracellular vesicles (sEVs) derived from lipopolysaccharide (LPS)-treated stem cells from the apical papilla (SCAP) in modulating macrophage polarization and osteoblast differentiation. SCAPs were treated with LPS for 24 h, and sEVs from untreated (SCAP-sEVs) and LPS-treated SCAP (LPS-SCAP-sEVs) were isolated via ultracentrifugation and characterized using transmission electron microscopy, Western blot, and Tunable Resistive Pulse Sensing. LPS-SCAP-sEVs exhibited characteristic exosome morphology (~100 nm diameter) and expressed vesicular markers (CD9, CD63, CD81, and HSP70). Functional analysis revealed that LPS-SCAP-sEVs promoted M1 macrophage polarization, as evidenced by the increased pro-inflammatory cytokines (IL-6 and IL-1β) and the reduced anti-inflammatory markers (IL-10 and CD206), while impairing the M2 phenotype. Additionally, LPS-SCAP-sEVs had a minimal impact on SCAP metabolic activity or osteogenic gene expression but significantly reduced mineralization capacity in osteogenic conditions. These findings suggest that sEVs mediate the inflammatory interplay between SCAP and macrophages, skewing macrophage polarization toward a pro-inflammatory state and hindering osteoblast differentiation. Understanding this sEV-driven communication axis provides novel insights into the cellular mechanisms underlying inflammation in oral tissues and highlights potential therapeutic targets for modulating extracellular vesicle activity during acute inflammatory episodes.

## 1. Introduction

Pulpitis, or inflammation of the dental pulp tissue, is a prevalent condition characterized by severe dental pain, and often resulting in the excessive use of analgesics and opioids [1]. The dental pulp is a complex tissue comprising various cell types, including immunocompetent cells, fibroblasts, stem cells, and odontoblasts. In the event of tissue injury, bacterial components trigger immune responses that involve a dynamic interplay between these cellular components. Macrophages are central players in this process, orchestrating the inflammatory response and tissue repair through their ability to transition from a pro-inflammatory (M1) to a regenerative (M2) phenotype as healing progresses [2,3]. Recent studies have revealed that macrophages in regenerating pulp tissue shift from an M1-dominant phenotype to an M2-dominant phenotype over time. This M1-to-M2 transition is crucial for creating a favorable microenvironment that supports pulp tissue regeneration [4,5]. The transition from M1 to M2 macrophages has been linked to enhanced odontogenic and osteogenic differentiation of dental pulp stem cells (DPSCs), facilitating the repair and restoration of damaged pulp tissue [6].

However, while traditional root canals and vital pulp therapies effectively address infection and inflammation, they fail to fully restore the biological functionality of the dental pulp. This limitation has driven a shift in focus toward regenerative therapies, with mesenchymal stem cells (MSCs) emerging as pivotal players in tissue engineering due to their multi-lineage differentiation capacity and immunomodulatory properties [7,8].

Tissue engineering strategies using cell-based approaches face numerous challenges, including the potential for immune rejection and limited access to suitable cell sources. Stem cells from the apical papilla (SCAP) have the ability to differentiate into primary odontoblasts and dental pulp cells. SCAPs have been shown to secrete anti-inflammatory cytokines, including IL-10, IL-4, and IL-13 as well as soluble factors and extracellular vesicles (EVs) [9,10,11]. Recent studies have demonstrated a close link between stem cells like SCAP and innate immunity cells, such as macrophages, which allows for the resolution of inflammation and a return to homeostasis [3,12,13,14].

‘Extracellular vesicles’ is a broad term that encompasses a variety of cell-derived vesicles released into the extracellular environment. EVs can be categorized into several subtypes, including small extracellular vesicles (sEVs) (often called exosomes), microvesicles, and apoptotic bodies that differ in terms of their biogenesis, size, and composition [15,16].

In recent years, sEVs have emerged as key mediators of cell-to-cell communication [17]. While the functions of sEVs in physiological conditions have been extensively studied, their characterization and biological roles in inflammatory contexts remain relatively unexplored. Understanding these roles is crucial for deciphering their impact on inflammatory processes. Consequently, our study seeks to characterize sEVs derived from lipopolysaccharide-treated SCAP (sEVs-LPS) and to investigate two aspects of their potential capacities: the autocrine effect of sEVs-LPS on the osteogenic differentiation potential of SCAP and their paracrine effects on macrophage polarization.

## 2. Results

### 2.1. Characterization of sEVs-CT and sEVs-LPS

sEVs-LPS were isolated and characterized from SCAPs cultured under inflammatory conditions induced by LPS stimulation and compared with control. The sizes of sEVs-CT and sEVs-LPS were similar, with diameters of approximately 94.7 ± 5.9 nm for sEVs-CT and 96 ± 5.1 nm for sEVs-LPS (Figure 1a–c). The concentration of sEVs-LPS was found to be twice as high as that of sEVs-CT (Figure 1d). Transmission Electron Microscopy examination revealed the ultrastructure with lipid layers, representing the plasma membrane of sEVs, with particle sizes around 100 nm for both types of sEVs, in adequation with TRPS measurement (Figure 1e). Interestingly, the protein quantity in sEVs-CT was higher compared to sEVs-LPS, with approximately 3 ng per million particles for sEVs-CT versus 1 ng for sEVs-LPS (Figure 1f). Furthermore, Western blot analysis confirmed the presence of vesicular markers in both sEVs-CT and sEVs-LPS. As expected, CD9, CD63, CD81, and HSP70, but not GM130 (a Golgi marker used as a negative control), were detected in sEVs (Figure 1g).

### 2.2. Effect of sEVs-LPS on SCAP Osteoblastic Differentiation and Metabolic Activity

As expected, the expression of osteogenic markers *ALP*, *Col1a1*, and *Runx2* increased during the differentiation process. However, no significant differences were observed when cells were treated with sEVs-CT or sEVs-LPS compared to the untreated control (Figure 2a–c).

In the mineralized nodule formation assay, strong calcium deposits were observed in the presence of conditioned medium from both LPS-treated and control SCAP. In contrast, sEVs derived from either the control or LPS-treated cells alone completely suppressed the mineralization capacity of SCAP (Figure 2d). The mineral deposits were five times lower in the presence of both sEVs-CT and sEVs-LPS than in the presence of CT-CM and LPS-CM (Figure 2e). Regarding the effect of sEVs on the metabolic activity of SCAP, no significant differences were observed between sEVs-LPS and sEVs-CT, regardless of the dose used, up to seven days (Figure 2f). Taken together, these findings suggest that sEVs-LPS reduce the bone regeneration potential of SCAP cells.

### 2.3. Effect of sEVs-LPS on Macrophages Polarization

The effect of sEVs at 3 doses (1X, 5X, and 10X) on M0, M1, and M2 macrophages was assessed. While no statistical differences were observed for TNF-α, the expression of key pro-inflammatory genes, *IL-6* and *IL-1β*, was significantly increased in M0 and M1 macrophages treated with sEVs-LPS compared to the untreated control (Figure 3a,c). Conversely, the expression of the anti-inflammatory genes, *IL-10* and *CD206*, significantly decreased in M2 macrophages in the presence of sEVs-LPS (Figure 3d,e). No significant effect of sEVs-CT on polarized macrophages was observed at any dose for *TNF-α* (Figure 3b). These results were further confirmed by a multiplex bead-based immune assay, which demonstrated that in an inflammatory context, sEVs-LPS exacerbated the pro-inflammatory phenotype of M0 macrophages (Figure 3f–h).

## 3. Discussion

The resolution of inflammation in pulpitis requires the modulation of excessive tissue inflammatory response to promote pulp repair or regeneration. In this study, we investigated the effect of sEVs derived from LPS-treated stem cells of the apical papilla on macrophage polarization and osteoblast differentiation to address this issue.

Our study provides a standard characterization of small extracellular vesicles, derived from SCAP, in line with the recommendations of the MISEV guidelines [18]. Several parameters may be modified to enhance the production of EVs by parent cells, such as adjusting the medium composition, oxygen levels, culture duration, or applying shear stress [19]. To align with clinical relevance, we chose to isolate extracellular vesicles from SCAPs treated with LPS stimulation for 24 h without applying mechanical stress or hypoxia. Although shear stress or complex forces (compression or tension) applied to parent cells can increase EVs, they may also alter their biological activity, cargo composition, and uptake by recipient cells. Excess debris resulting from biomechanical forces during EVs production may pose a significant issue for EV purity and, potentially, impact their activity [20].

The observed increase in vesicle concentration following LPS stimulation suggests that inflammatory stress enhances exosome biogenesis, potentially driven by upregulated pathways associated with cellular stress responses. This increase in vesicle concentration under LPS-induced conditions aligns with evidence from other inflammatory models, indicating that heightened cellular stress can stimulate exosome production, which may play a role in modulating the inflammatory microenvironment [21].

Additional characterization elements are required, which will be essential aspects of our future investigations. Moreover, it would be relevant to study the uptake of sEVs derived from SCAP by macrophages as the mechanisms of internalization remain poorly understood but are crucial for clinical application [22].

Although sEVs-LPS were found to contain three times less protein than sEVs-CT, they still significantly influenced the production of IL-6 and IL-1β. The relationship between protein content and cytokine secretion suggests that specific proteins driving these effects may differ, though this has yet to be determined. It would be valuable to first conduct a proteomic analysis to investigate the inflammatory pathways, including cytokines present in EVs or known inducers of expression. This should be performed prior to performing next-generation sequencing (NGS) to identify differences in SCAP sEV miRNAs. This approach would allow for the identification of miRNAs that are either enriched or depleted in sEVs-LPS compared to sEVs-CT.

Macrophages play a crucial role in resolving inflammation. Understanding the activation of pro-inflammatory macrophages by oral stem cells and their secretome is essential for controlling the initial phases of the immune response during acute inflammation. Our study demonstrates that sEVs-LPS exacerbate the pro-inflammatory phenotype of M1 macrophages. This finding underscores the potential crosstalk between oral stem cells and macrophages in the early stages of inflammation via sEVs. Previous studies have demonstrated that M2-polarized macrophages enhance osteoblast differentiation through the secretion of pro-regenerative cytokines, such as IL-10 and TGF-β, which promote an anti-inflammatory and osteogenic microenvironment, while M1 macrophages suppress osteogenesis via pro-inflammatory mediators, including TNF-α and IL-6 [23]. The skewing of macrophages toward an M1 phenotype by sEVs-LPS could have significant downstream implications for osteoblast activity and bone regeneration potential. Crucially, the uptake of sEVs by macrophages is pivotal in this process, enabling the delivery of bioactive molecules that modulate macrophage behavior and function (Appendix A).

Although sEVs from untreated stem cells are known for their anti-inflammatory, immunomodulatory, angiogenic, and pro-regenerative properties [24,25,26] in various pathological contexts, sEVs derived from LPS-treated cells (sEVs-LPS) have been shown to exacerbate M1 macrophage phenotypes. For example, Kang et al. (2018) already demonstrated that EVs derived from LPS pre-conditioned periodontal ligament stem cells induce M1 macrophage polarization [27]. Similarly, in kidney injury, sEVs from tubular epithelial cells also promote M1 polarization through miRNA-19b-3p [28]. Furthermore, Carceller et al. (2021) found that the anti-inflammatory effect of mouse-derived adipose tissue mesenchymal stem cell secretome is mediated primarily by soluble factors, rather than EVs. This insight sheds light on the downregulation of TLR4 expression and NF-κB pathway mediated solely by soluble factors during the early stages of inflammation [29]. Our research supports these findings, demonstrating that EVs-LPS enhance the pro-inflammatory macrophage phenotype. While we used the THP-1 monocytic cell line, which may not fully represent the complexity of macrophage phenotypes in vivo, our study provides valuable insights into the effects of sEVs on polarized macrophages in an inflammatory context.

Notably, our findings suggest that through their effect on osteoblastic cells, sEVs-LPS may reduce the bone regeneration potential of SCAPs. Contradictory results have been obtained with sEVs from other cells, such as dental follicle cells treated with LPS on periodontal ligament cells from patients with periodontitis [30]. In our study, we tested the potential of our EVs on SCAP from healthy patients. The apical papilla is the precursor tissue of the radicular pulp, and SCAPs interact with dental pulp stem cells and odontoblastic cells. Therefore, it is crucial to assess how sEVs can impact the pro-regenerative properties of SCAP in cases of injury. In 2021, Nakao et al. demonstrated that sEVs derived from TNF-α treated human gingiva mesenchymal stem cells inhibit periodontal bone loss in a ligature-induced periodontal model in mice [31].

Although this study presents novel insights into the role of sEVs derived from LPS-treated SCAP in modulating macrophage polarization and osteoblast differentiation, several limitations should be acknowledged. Firstly, the findings are based on in vitro experiments, which, while providing valuable mechanistic insights, may not fully replicate the complexity of the in vivo oral microenvironment. Future investigations employing animal models or clinical samples are necessary to validate the translational relevance of these results. Secondly, the use of the THP-1 monocytic cell line for macrophage polarization experiments, though well-established, does not fully capture the phenotypic and functional diversity of primary macrophages, which could provide a more comprehensive understanding. Additionally, while the study confirmed vesicular markers in sEVs and demonstrated their functional effects, detailed proteomic and miRNA analyses were not conducted. Such analyses could reveal the specific bioactive molecules responsible for the observed effects and further elucidate their mechanisms of action. The study also focuses on acute inflammatory conditions induced by LPS, which may not fully represent chronic or low-grade inflammation scenarios often encountered in clinical settings. Lastly, while the suppression of osteoblast mineralization capacity by sEVs was observed, the underlying molecular pathways remain to be elucidated, warranting further exploration in future studies.

Although this study presents novel insights into the role of sEVs derived from LPS-treated SCAP in modulating macrophage polarization and osteoblast differentiation, several limitations should be acknowledged. Firstly, the findings are based on in vitro experiments, which may not fully replicate the complexity of the in vivo oral microenvironment. Future investigations employing animal models or clinical samples are necessary to validate the translational relevance of these results. Moreover, the small sample size, which, although sufficient for demonstrating significant trends, may reduce the generalizability of the findings. Future studies with larger sample sizes are needed to validate these results and enhance statistical power. Secondly, the use of the THP-1 monocytic cell line for macrophage polarization experiments does not fully capture the phenotypic and functional diversity of primary macrophages, which could provide a more comprehensive understanding. Additionally, while the study confirmed vesicular markers in sEVs and demonstrated their functional effects, detailed proteomic and miRNA analyses were not conducted. Such analyses could reveal the specific bioactive molecules responsible for the observed effects and further elucidate their mechanisms of action. The study also focuses on acute inflammatory conditions induced by LPS, which may not fully represent chronic or low-grade inflammation scenarios often encountered in clinical settings. Lastly, while the suppression of osteoblast mineralization capacity by sEVs was observed, the underlying molecular pathways remain to be elucidated, warranting further exploration in future studies.

## 4. Materials and Methods

### 4.1. Isolation of sEVs Derived from SCAP

Apical papilla tissue was collected from third molars extracted from medically healthy patients aged 16 to 25 years. The SCAP culture was established as previously described [32]. Cells were cultured in the alpha minimum essential medium (α-MEM) (Eurobio, Les Ulis, France), and supplemented with penicillin (1 U/mL, Gibco, Grand Island, NY, USA), streptomycin (1 µg/mL, Gibco), and L-glutamine (2 mM, Gibco) and 10% fetal bovine serum (Dutscher, Brumath, France). When the SCAPs (3rd to 5th passage) reached 80% confluence, they were washed with phosphate-buffered saline (PBS) and cultured in a medium containing exosome-depleted serum. SCAPs were treated with LPS from *E. coli* O55:B5 (Sigma-Aldrich, St. Louis, MO, USA) at 1 µg/mL for 24 h. The supernatant was collected and centrifuged at 300× *g* for 10 min (to eliminate cell debris), followed by centrifugation at 10,000× *g* for 30 min (to remove large vesicles). The resulting supernatant was then centrifuged at 100,000× *g* for 70 min. The pellet was washed in cold PBS and centrifuged again at 100,000× *g* for another 70 min. After the final centrifugation, the pellet was resuspended in cold PBS. All centrifugations were performed at 4 °C.

### 4.2. Scanning Transmission Electron Microscopy

The morphology of sEVs derived from SCAP was analyzed using a transmission electron microscope (SEM GeminiSEM300 (STEM detector) Zeiss, Oberkochen, Germany). Briefly, sEVs were fixed with a 2% paraformaldehyde solution. A suspension (3 × 10^8^ sEVs) of sEVs was deposited on the hydrophilic surface of formvar carbon-coated copper 200 mesh grids (AGS162 Agar Scientific, Rotherham, UK), followed by a 20 min adsorption period. The grid was transferred to a droplet of PBS for washing and then to a droplet of 1% glutaraldehyde for 5 min. The grid underwent eight washes with PBS immersed in Uranyless for 1 min for contrast enhancement. After this process, the grid was air-dried for 24 h before being subjected to microscopic analysis. The grids were coated with a thin layer of platinum (3 nm) using a Leica EM ACE600 high-vacuum sputter coater (Leica Microsystems, Wetzlar, Germany). Grids with negatively stained vesicles were observed with a GeminiSEM 300 Zeiss scanning electron microscope, equipped with a STEM detector (Carl Zeiss AG, Oberkochen, Germany). Observations were made at 29 keV, 7.5 µm diaphragm, and a working distance of 4 mm.

### 4.3. Small Extracellular Vesicles Size and Concentration Measurement

The size and concentration of sEVs derived from SCAP were assessed using Tunable Resistive Pulse Sensing (TRPS) technology, with an Exoïd instrument (Izon, Lyon, France). Isolated sEVs were diluted in PBS and loaded into a 150 nm nanopore (NP150, Izon, Lyon, France), and Exoïd conducted size and concentration measurements for each sample at three different pressures. Calibration particles (CPC200, Izon, Lyon, France) were used as references to determine size and concentration. The corresponding Izon Data Suite software V3.4 was used to analyze the collected data.

### 4.4. Western Blot

The protein concentration of sEVs was determined using a microBCA assay kit (Thermo Fisher, Rockford, IL, USA). sEVs derived from SCAPs were incubated on ice with a RIPA lysis buffer (Santa Cruz Biotechnology, Santa Cruz, CA, USA), and diluted tenfold in accordance with the manufacturer’s instructions.

For Western blot analysis, 2 µg of sEVs were loaded into stain-free, 4–15% polyacrylamide gels (Bio-Rad, Hercules, CA, USA) and separated via gel electrophoresis. Subsequently, the proteins were then transferred to a membrane using a transblot system and probed with antibodies against CD9 (1:1000), CD63 (1:1000), CD81 (1:1000), HSP 70 (1:1000), and GM130 (1:1000), which were used as a negative control. The secondary antibody was diluted to a 1:20,000 ratio in Tris Buffered Saline with 0.1% Tween. All antibodies, except for GM130 (Abcam, Catalog No. 52649, Cambridge, UK), were obtained from SBI-Ozyme (ExoAb KIT, SBI, Palo Alto, CA, USA). Protein bands were detected using a ChemiDoc system and analyzed with Image Lab 4.1 software (Bio-Rad).

### 4.5. Metabolic Activity Assay

SCAPs were initially seeded at a density of 1 × 10^4^ cells per well in a 96-well plate, using the α-MEM culture medium, supplemented with 10% exosome-depleted fetal bovine serum (Dutscher, Brumath, France). This medium was further supplemented with penicillin (1 U/mL, Gibco), streptomycin (1 µg/mL, Gibco), and L-glutamine (2 mM, Gibco).

Upon reaching 80% confluence, a CCK-8 assay (Cell Counting Kit-8, Sigma-Aldrich, St. Louis, MO, USA) was conducted on day 0 (D0) to establish the initial metabolic activity of the cells in the absence of treatment. Briefly, the cells were rinsed with 200 µL of PBS and replenished with 200 µL of fresh exosome-free medium. Subsequently, 20 µL of CCK-8 reagent was added to each well, followed by a 1 h incubation at 37 °C. At this point, 100 µL of the supernatant was transferred to a 96-well flat-bottom plate, and the absorbance was measured at 450 nm using a microplate reader (Tristar, Berthold Technologies, Bad Wildbas, Germany). On subsequent days (D3, D5, and D7), the cells were either stimulated or left untreated, with the addition of sEVs derived from SCAP or SCAP induced by lipopolysaccharide (LPS), at two different doses, 1X (1 × 10^7^ sEVs per well) and 5X (5 × 10^7^ sEVs per well). The measurement of metabolic activity was carried out in a similar manner for each time point after treatment.

### 4.6. SCAP Osteoblastic Differentiation

SCAPs were seeded at 1 × 10^5^ cells per well in 12-well plates in the α-MEM culture medium supplemented with 10% exosome-depleted fetal bovine serum (Dutscher) and supplemented with penicillin (1 U/mL, Gibco), streptomycin (1 µg/mL, Gibco), and L-glutamine (2 mM, Gibco). Each time point was allocated a separate plate for seeding. Upon reaching 80% confluence, osteogenic differentiation was induced by supplementing the culture with ascorbic acid (50 µM), β-glycerophosphate (10 mM), and dexamethasone (100 nM), all purchased from Sigma-Aldrich, in 10% exosome-depleted fetal bovine serum. On day 0, the cells were treated with sEVs (5 × 10^8^ sEVs per well) on day 0. The medium containing the inducers and sEVs was refreshed on day 3. Total RNA was extracted on days 0, 1, 3, and 6.

For the evaluation of cellular calcium deposition, Alizarin Red S staining was performed in 24-well plates under the same cell culture conditions. The osteogenic medium was refreshed twice a week, accompanied by treatment with sEVs-CT or sEVs-LPS (2.5 × 10^8^ sEVs per well) or conditioned medium from SCAP treated or not with LPS). On day 17, the cells were washed with PBS and stained with 40 mM Alizarin Red S (Sigma-Aldrich) for 20 min. After rinsing the cells three times with ultrapure water, they were imaged using a scanner. To dissolve the mineralized crystals, a solution of 20% methanol and 10% acetic acid in water was applied for 15 min (400 µL per well). Finally, the supernatant was transferred to a 96-well flat-bottom plate, and the optical density was measured at 450 nm using a microplate reader.

### 4.7. Macrophages Polarization

Human monocytic THP-1 cells (ATCC, Manassas, VA, USA) were seeded in 12-well plates at a density of 7.5 × 10^5^ cells per well. They were cultured in the Roswell Park Memorial Institute (RPMI) medium (Gibco), supplemented with 10% exosome-depleted fetal bovine serum (Dutscher), penicillin (1 U/mL, Gibco) and streptomycin (1 µg/mL, Gibco).

To differentiate THP-1 monocytes into M0 macrophages, THP-1 cells were initially incubated with 100 nM phorbol 12-myristate 13-acetate (Sigma-Aldrich) for 24 h, followed by 48 h in a RPMI medium. Macrophages were then polarized into M1 macrophages by incubating them for 24 h with 20 ng/mL of human recombinant Interferon-gamma (Biolegend, San Diego, CA, USA) and 10 pg/mL of Lipopolysaccharide (LPS) obtained from *E. coli* O55:B5 (Sigma-Aldrich). M2 macrophage polarization was achieved by incubating the cells for 72 h with 20 ng/mL of recombinant human Interleukin 4 (Biolegend) and 20 ng/mL of recombinant human Interleukin 13 (Biolegend). Following polarization, sEVs derived from SCAP were applied to the cells at three different doses: 1X (5 × 10^8^ sEVs per well), 2X (1 × 10^9^ sEVs per well), and 10X (5 × 10^9^ sEVs per well) for a 24 h incubation period. Viability tests performed prior to the application of sEVs indicated that the macrophages were healthy and polarized correctly under the experimental conditions (Appendix A).

### 4.8. Quantitative Real-Time PCR (RT-qPCR)

To assess the influence of sEVs derived from SCAP on both SCAP osteoblastic differentiation and macrophage polarization, RT-qPCR was performed. Total RNA extraction was extracted using the NucleoSpin RNA extraction kit (Macherey-Nagel, Düren, Germany), according to the manufacturer’s instructions. Subsequently, 1 µg of total RNA was reverse transcribed using the Verso cDNA synthesis kit (ThermoFisher Scientific, Saint-Aubin, France). Real-time PCR was conducted with the SyBR Select Master Mix (Applied Biosystems, Foster City, CA, USA) and analyzed on a Bio-Rad CFX96 detection system (Bio-Rad). The cycling conditions were initial denaturation at 95 °C for 2 min, followed by 40 cycles of denaturation at 95 °C for 15 s, and annealing/extension at 60 °C for 30 s. The specificity of the amplified products was confirmed by analyzing the melting curves. mRNA expression levels were normalized to housekeeping genes (*18S* and *PPIA*), and calculations were carried out using the 2^−ΔΔ^ CT method, with non-stimulated M0 cells serving as the control. The primers were provided by Eurofins (MWG Operon, Nantes, France) (Table 1).

### 4.9. Bead Based Immunoassay

A bead-based multiplex assay (LEGENDplex™ human macrophages/microglia panel, Biolegend, San Diego, CA, USA) was used to measure cytokines secretion following macrophage polarization. The assay was performed according to the manufacturer’s instructions. Briefly, detection beads were vortexed and incubated with a supernatant of polarized THP-1 cells. Beads bound to target cytokines were then washed, incubated with detection antibodies, and then washed again, before incubated with streptavidin–phycoerythrin, which binds to the biotinylated detection antibodies; the beads were then washed once more. Cytokine concentrations were then determined through flow cytometry analysis. This assay allows for the simultaneous quantification of 10 key targets involved in monocyte differentiation and macrophage functions, including IL-10, IL-12p70, IL-12p40, IL-1RA, IL-1β, IL-23, IL-6, IP-10, TARC, and TNF-α. Data were analyzed using Legendplex™ software version 7.0 and reported as pg/mL.

### 4.10. Statistical Analysis

All statistical analyses were performed using one-way analysis of variance (ANOVA), followed by Tukey’s multiple comparison test to assess significant differences. However, for the analysis of size distribution, concentration, and protein concentrations of sEVs, multiple two-tailed Student’s *t*-tests, without correction for multiple comparisons, were employed. Statistical analyses were conducted using GraphPad Prism v10.00 (GraphPad Software, San Diego, USA). Data are presented as the mean ± SEM (standard error of the mean). For all assays, a minimum of three independent experiments was performed. The significance of *p*-values was determined as follows: * *p* < 0.05, ** *p* < 0.01, and **** *p* < 0.0001.

## 5. Conclusions

In conclusion, our research underscores the importance of understanding the specific roles and underlying mechanisms of sEVs in inflammation and regeneration. Future studies should focus on further characterization of sEVs and their cargo, including miRNA content, to identify therapeutic targets for modulating inflammatory responses and enhancing tissue regeneration. An elegant therapeutic strategy could involve the use of sEVs as vectors for the targeted delivery of specific miRNA or protein inhibitors or activators, potentially affecting other biological processes beyond inflammation. Additionally, another promising approach could involve the use of inhibitors of sEVs biogenesis in the early stages of inflammation [33,34] to mitigate their pro-inflammatory effects. This could potentially lead to novel treatments for pulpitis leveraging the unique properties of sEVs derived from oral stem cells.

## Figures and Tables

**Figure 1 ijms-26-00297-f001:**
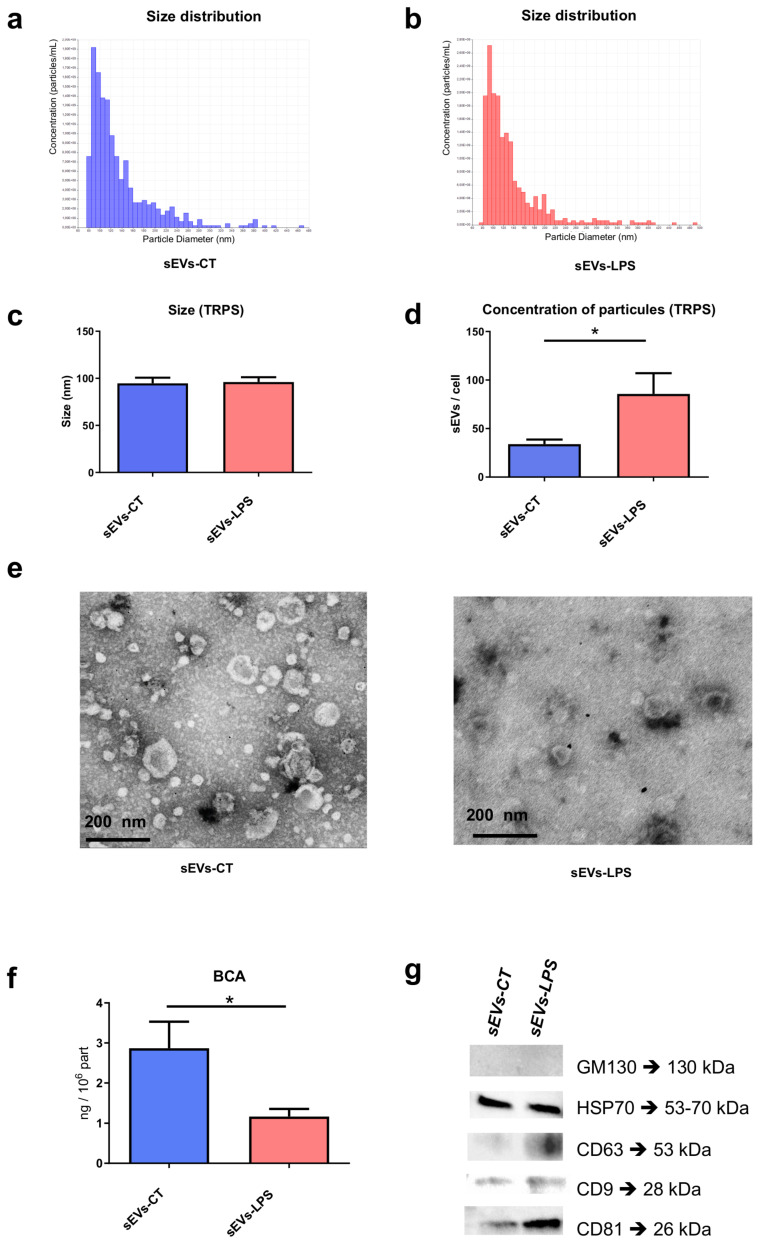
Characterization of sEVs derived from control SCAP (sEVs-CT) (**a**) or from LPS-treated SCAP (sEVs-LPS) (**b**). Size distribution and concentration of sEVs-CT (**a**) and sEVs-LPS (**b**) assessed using Tunable Resistive Pulse Sensing (TRPS) technology. (**c**) TRPS analysis showing the mean size of sEVs-CT and sEVs-LPS. (**d**) TRPS analysis showing the quantity of sEVs-CT and sEVs-LPS secreted per cell. (**e**) Ultrastructure of sEVs-CT and sEVs-LPS observed using electron microscopy. (**f**) Protein concentrations of sEVs-CT and sEVs-LPS determined by micro-BCA assay. (**g**) Expression of sEV-associated protein positive markers (HSP70, CD63, CD9, and CD81). GM130, a cis-Golgi matrix protein, was used as a negative control to confirm the absence of Golgi apparatus contamination by Western blot. Data are expressed as mean ± SEM. *N* = 3. * *p* < 0.05 (Student’s *t*-test).

**Figure 2 ijms-26-00297-f002:**
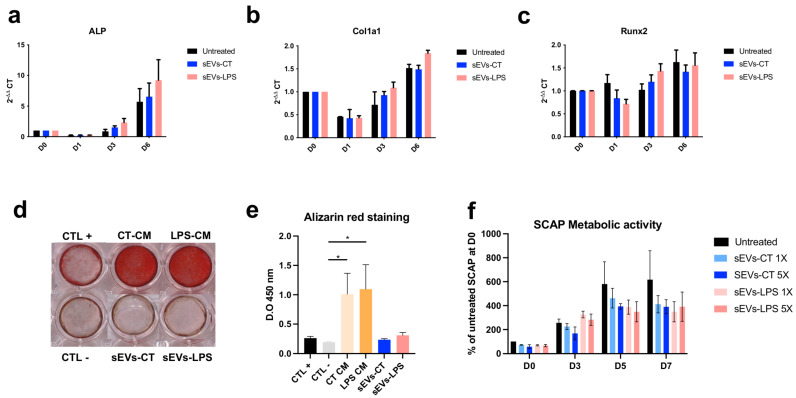
Effect of sEVs-CT and sEVs-LPS on SCAP osteoblastic differentiation and mineralization capacity. (**a**–**c**) Expression of representative genes of osteoblastic differentiation, including *ALP* (**a**)*, Col1a1* (**b**), and *Runx2* (**c**), determined by RT-qPCR. Gene expression (2^−ΔΔ^ CT) was normalized to SCAP at day 0 and presented as fold change. In total, *18S* and *PPIA* were used as housekeeping genes. (**d**) Alizarin Red S staining performed on SCAP cultured in an osteogenic induction medium for 17 days. The effects of untreated and LPS-treated SCAP conditioned medium were compared. (CT +: positive control (osteogenic medium); CT −: negative control (the osteogenic medium with exosome-depleted fetal bovine serum); CT-CM: Conditioned medium; LPS-CM: Conditioned medium with LPS). (**e**) Quantification of the degree of mineralization assessed by dissolution of mineralization crystals with methanol and acetic acid. The optical density was measured at 450 nm. (**f**) Effect of 1X (1 × 10^7^ sEVs per well) and 5X (5 × 10^7^ sEVs per well) of sEVs-CT and sEVs-LPS on SCAP metabolic activity determined by CCK-8 assay. The results are presented as the percentage of metabolic activity relative to untreated SCAP at day 0. The data are expressed as mean ± SEM from three independent experiments (*N* = 3), each performed in triplicate (*n* = 3). *p*-values derived from one-way ANOVA followed by Tukey’s multiple comparison tests is * *p* < 0.05.

**Figure 3 ijms-26-00297-f003:**
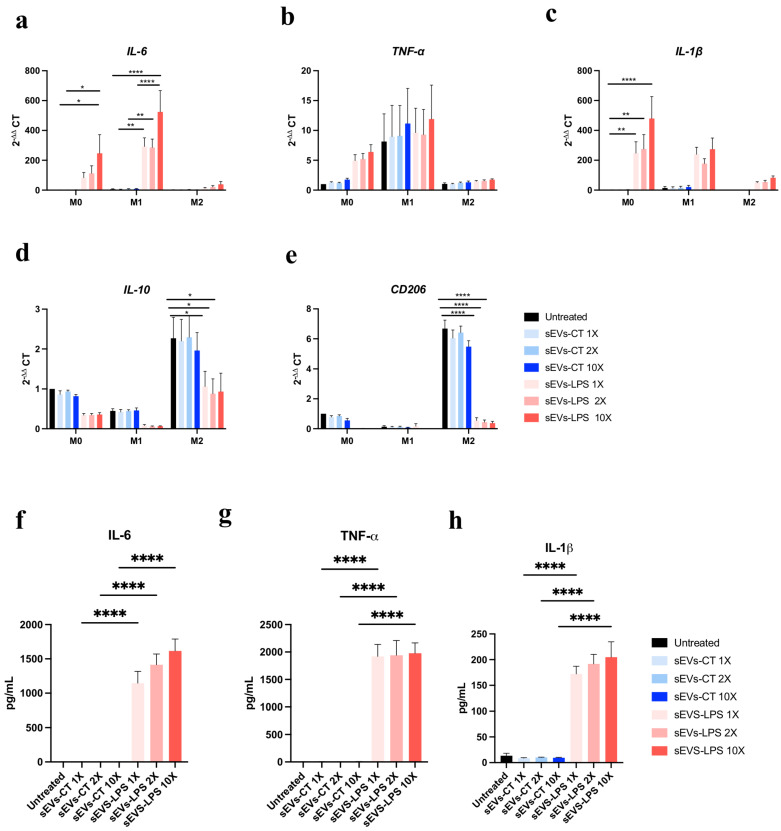
Effect of sEVs-LPS on polarized macrophages. Expression of representative pro-inflammatory marker genes including *IL-6* (**a**), *TNF-α* (**b**), and *IL-1β* (**c**); and anti-inflammatory marker genes, *IL-10* (**d**) and *CD206* (**e**), after treatment of polarized macrophages with sEVs derived from SCAP or LPS-induced SCAP for 24 h determined by RTqPCR. Gene expression (2^−ΔΔ^ CT) was normalized to untreated M0 macrophages and presented as fold change. The *18S* and *PPIA* were used as housekeeping genes. Quantification of pro-inflammatory factors IL-6 (**f**), TNF-α (**g**), and IL-1β (**h**) in the culture supernatant of M0 macrophages using bead-based multiplex assays by flow cytometry. The data are expressed as mean ± SEM. *N* = 3; *n* = 3. *p*-values derived from one-way ANOVA followed by Tukey’s multiple comparison tests are * *p* < 0.05, ** *p* < 0.01, and **** *p* < 0.0001.

**Table 1 ijms-26-00297-t001:** RT-qPCR Primer sequences.

Target	Forward 5′ => 3′	Reverse 5′ => 3′
*Col1a1*	CTC CTG ACG CAC GGC C	CCG TTC TGT ACG CAG GTG ATT
*ALP*	GGG ATA AAG CAG GTC TTG GGG TGC	CGC TTG GTC TCG CCA GTA CTT GG
*Runx2*	CCT AAA TCA CTG AGG CGG TC	CAG TAG ATG GAC CTC GGG AA
*IL-6*	TCC ACA AGC GCC TTC GGT CCA G	CTC AGG GCT GAG ATG CCG TCG
*TNFα*	CAG CCT CTT CTC CTT CCT GAT	GCC AGA GGG CTG ATT AGA GA
*IL-1β*	CCG GGA CTC ACA GCA AAA	GGA CAT GGA GAA CAC CAC TTG
*IL-10*	TAC GGC GCT GTC ATC GAT TT	TAG AGT CGC CAC CCT GAT GT
*CD206*	AGC CAA CAC CAG CTC CTC AAG A	CAA AAC GCT CGC GCA TTG TCC A
*18S*	AGC AAA CCC CAA CTC AAC C	GTC CCT CAG AAG GGG TGA C
*PPIA*	GTC AAC CCC ACC GTG TTC TT	CTG CTG TCT TTG GGA CCT TGT

## Data Availability

The datasets and materials used and/or analyzed during the current study are available from the corresponding author upon reasonable request.

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
