# Peer review of "Small Extracellular Vesicles Derived from Lipopolysaccharide-Treated Stem Cells from the Apical Papilla Modulate Macrophage Phenotypes and Inflammatory Interactions in Pulpal and Periodontal Tissues"

_ijms, 2024, doi:10.3390/ijms26010297_

Round 1

Reviewer 1 Report

Comments and Suggestions for Authors

The manuscript contains original and important principles, interesting and relevant information. The study was quite well done and provided interesting data. The whole idea is clear. The Authors investigated the role of small extracellular vesicles (sEVs) derived from lipopolysaccharide (LPS)-treated stem cells from the apical papilla (SCAP) in modulating macrophage polarization and osteoblast differentiation. The chosen topic is very interesting and important.  Results presented by the Authors are interesting and valuable. Findings of the present study potentially may have a significant impact for further studies trying to understand and explain the cellular mechanisms underlying inflammation in oral tissues and to identify therapeutic targets for modulating inflammatory responses and enhancing tissue regeneration. The section of materials and methods is quite clear described, presented and understandable. Results are clearly presented, the provided figures are understandable and support the main findings of this study. The abstract clearly describes the content of the article and importance of the chosen topic. Provided references are relevant to the presented topic. Conclusions are consistent with the arguments presented in the study.

The remark regarding this paper:

1) Authors did not mention what are the limitations of the work. Please provide the missing information.

This paper is good and after the minor revision of mentioned aspect, the work might be published.

Author Response

Comment: Authors did not mention what are the limitations of the work. Please provide the missing information.

Response: We thank the reviewer for their thoughtful feedback and positive assessment of our manuscript. We acknowledge the importance of addressing the limitations of the study to provide a balanced perspective and identify avenues for future research. A paragraph discussing the limitations of the study has been added to the manuscript in the revised discussion section for clarity and transparency. We hope this revision meets the expectations of the reviewer.

Added paragraph: "Although this study presents novel insights into the role of sEVs derived from LPS-treated SCAP in modulating macrophage polarization and osteoblast differentiation, several limitations should be acknowledged. Firstly, the findings are based on in vitro experiments, which, while providing valuable mechanistic insights, may not fully replicate the complexity of the in vivo oral microenvironment. Future investigations employing animal models or clinical samples are necessary to validate the translational relevance of these results. Secondly, the use of the THP-1 monocytic cell line for macrophage polarization experiments, though well-established, does not fully capture the phenotypic and functional diversity of primary macrophages, which could provide a more comprehensive understanding. Additionally, while the study confirmed vesicular markers in sEVs and demonstrated their functional effects, detailed proteomic and miRNA analyses were not conducted. Such analyses could reveal the specific bioactive molecules responsible for the observed effects and further elucidate their mechanisms of action. The study also focuses on acute inflammatory conditions induced by LPS, which may not fully represent chronic or low-grade inflammation scenarios often encountered in clinical settings. Lastly, while the suppression of osteoblast mineralization capacity by sEVs was observed, the underlying molecular pathways remain to be elucidated, warranting further exploration in future studies."

Reviewer 2 Report

Comments and Suggestions for Authors

In this paper, the authors try to characterize SEVs-LPS and to investigate their autocrine effect on the osteogenic differentiation potential and their paracrine effects on macrophage polarization. Some aspects of this paper need to be improved in order to be accepted for publication:

Major comments :

1) In general the quality of the figures need to be improved. Figure 1 , figure 2 and figure S1, the quality is poor and we can't barely see the letters, numbers, legends... it's all blurry.

2) I can’t seem to find the original western blots for all the proteins tested. Please specify in the full membranes provided to the journal the protein checked and the band size. Also in figure 1g, the control (GAPDH, tubulin) …is missing. 

Minor points:

1)  Is it possible to add on Figure 3 some original images that represent how M0, M1 and M2 macrophages look like ?

2) In figure 2 legend,  n=3 appears twice (line 126).

3) Please clarify in the discussion the limitations of your small n

Author Response

Major Comments:

Comment 1: In general the quality of the figures need to be improved. Figure 1 , figure 2 and figure S1, the quality is poor and we can't barely see the letters, numbers, legends... it's all blurry.

Response 1: Thank you for pointing out concerns regarding the figure quality. The reduced clarity may be due to the PDF conversion process, as the quality of the figures is satisfactory in the Word document used for submission. However, we have verified and ensured that all figures in the revised manuscript maintain high resolution for optimal readability.

Comment 2:  I can’t seem to find the original western blots for all the proteins tested. Please specify in the full membranes provided to the journal the protein checked and the band size. Also in figure 1g, the control (GAPDH, tubulin) …is missing. 

Response 2: We confirm that the original western blot membranes, clearly indicating the tested proteins and their corresponding band sizes, have already been submitted to the journal. For Figure 1g, GM130 was used as a negative control for Golgi contamination, as per standard practice. We appreciate your observation and have clarified this further in the revised figure legend.

Minor Comments

Comment 1: Is it possible to add on Figure 3 some original images that represent how M0, M1 and M2 macrophages look like ?

Response 1: Unfortunately, we do not have representative images of M0, M1, and M2 macrophages to include in Figure 3. Direct observation of the macrophages under the conditions tested did not reveal any abnormalities. While we do have images of M0 and M1 macrophages before the application of sEVs, we do not have images after treatment. However, viability tests performed prior to the application of sEVs demonstrated good results, indicating that the macrophages were healthy and polarized correctly under the experimental conditions. This information has been highlighted in the supplementary materials (Figure S2).

Comment 2: In figure 2 legend,  n=3 appears twice (line 126).

Response 2: Thank you for pointing this out. We would like to clarify that "N" and "n" have distinct meanings in our manuscript. "N = 3" refers to the number of independent experiments conducted, while "n = 3" indicates the number of technical replicates within each experiment. To avoid confusion, we have updated the legend in Figure 2 to explicitly define these terms for clarity. The revised legend now reads: "The data are expressed as mean ± SEM from three independent experiments (N = 3), each performed in triplicate (n = 3). P-values derived from one-way ANOVA followed by Tukey’s multiple comparison tests is *p < 0.05."

Comment 3: Please clarify in the discussion the limitations of your small n

Response 3: We have expanded the Discussion to address the limitations posed by the small sample size. While the findings are robust within the constraints of our experimental setup, we acknowledge that larger sample sizes would strengthen the generalizability of our conclusions. here is the added sentence; "

Moreover, the small sample size, which, although sufficient for demonstrating significant trends, may reduce the generalizability of the findings. Future studies with larger sample sizes are needed to validate these results and enhance statistical power".

Reviewer 3 Report

Comments and Suggestions for Authors

In this manuscript, the authors explored the role of small extracellular vesicles (sEVs) derived from lipopolysaccharide (LPS)-treated stem cells from the apical papilla (SCAP) in modulating macrophage polarization and osteoblast differentiation. The effects of cell polarization and osteoblast differentiation were also investigated. However, so many key points were missing in this manuscript. Therefore, I recommend that these essential experiments must be conducted and the results should be integrated into the manuscript before considering its acceptance for publication.

Major issues should be addressed.

1.     The authors wrote in the title that it is bone tissue related, but the manuscript is modeling the disease as pulpitis, so it is recommended that the authors revise the title to highlight periodontal disease.

2.     After LPS stimulation, the concentration of cell-secreted exosome vesicles became higher, and the authors were asked to discuss this phenomenon.

3.     The clarity of all the plots in Figure 2 could be improved.

4.     In section 2.2, no experimental results related to metabolism were observed, and the authors were requested to supplement this.

5.     What is the relationship between macrophage polarization and osteoblast differentiation, while no experimental validation of their interaction has been identified, and the authors are invited to elaborate on it and supplement the relevant experiments.

Comments on the Quality of English Language

The manuscript should be carefully checked and revised to avoid the spelling, expression and grammar errors. 

Author Response

We thank the reviewer for their detailed feedback and valuable suggestions. Below, we address each comment point by point.

Comment 1: The authors wrote in the title that it is bone tissue related, but the manuscript is modeling the disease as pulpitis, so it is recommended that the authors revise the title to highlight periodontal disease.

Response 1: We agree with the suggestion. The title has been revised to "Small extracellular vesicles derived from lipopolysaccharide-treated stem cells from the apical papilla modulate macrophage phenotypes and inflammatory interactions in pulpal and periodontal Tissues" to better reflect the manuscript's scope.

Comment 2:  After LPS stimulation, the concentration of cell-secreted exosome vesicles became higher, and the authors were asked to discuss this phenomenon.

Response 2: In line with the reviewer's suggestion, we expanded the discussion. "

The observed increase in vesicle concentration following LPS stimulation suggests that inflammatory stress enhances exosome biogenesis, potentially driven by upregulated pathways associated with cellular stress responses. This increase in vesicle concentration under LPS-induced conditions aligns with evidence from other inflammatory models, indicating that heightened cellular stress can stimulate exosome production, which may play a role in modulating the inflammatory microenvironment [21]."

Comment 3: The clarity of all the plots in Figure 2 could be improved.

Response 3: Thank you for pointing out concerns regarding the figure quality. The reduced clarity may be due to the PDF conversion process, as the quality of the figures is satisfactory in the Word document used for submission. However, we have verified and ensured that all figures in the revised manuscript maintain high resolution for optimal readability.

Comment 4: In section 2.2, no experimental results related to metabolism were observed, and the authors were requested to supplement this.

Response 4: We appreciate the reviewer’s observation regarding the lack of experimental results related to metabolism in Section 2.2. Unfortunately, we currently do not have additional experimental data addressing this aspect. However, the metabolic activity of SCAP treated with SEVs-LPS and SEVs-CT was evaluated using the CCK-8 assay, as presented in Figure 2f. These results indicated no significant differences in metabolic activity under the tested conditions, which aligns with our focus on the osteogenic differentiation and inflammatory modulation effects of the extracellular vesicles. We acknowledge the importance of investigating metabolic changes further and agree that a more detailed metabolic analysis could provide additional insights. This will be a priority in our future work to comprehensively understand the interplay between metabolism, osteogenesis, and macrophage modulation. We have updated the manuscript to emphasize this limitation and outline it as a perspective for future research.

Comment 5: What is the relationship between macrophage polarization and osteoblast differentiation, while no experimental validation of their interaction has been identified, and the authors are invited to elaborate on it and supplement the relevant experiments.

Response 5: We appreciate the reviewer’s comment regarding the relationship between macrophage polarization and osteoblast differentiation. While our current study focuses on the separate effects of SEVs-LPS on macrophage polarization and osteoblast differentiation, we acknowledge that experimental validation of their direct interaction would strengthen the manuscript.

Although we do not have direct experimental data to demonstrate this interaction, existing literature supports a close relationship between these processes. For instance, M2-polarized macrophages have been shown to enhance osteoblast differentiation through the secretion of pro-regenerative cytokines such as IL-10 and TGF-β, while M1 macrophages can suppress osteogenesis via pro-inflammatory mediators like TNF-α and IL-6. These findings align with the modulation of macrophage phenotypes observed in our study and their potential downstream impact on osteoblast activity. To address this limitation, we have expanded the discussion in the manuscript to include relevant references and highlight the need for future studies directly validating this interaction in the context of SEVs. This will be a key focus of our subsequent research efforts. We hope this clarification and the additional context adequately address the reviewer's concern.

Round 2

Reviewer 2 Report

Comments and Suggestions for Authors

All of my comments have been addressed by the authors

Reviewer 3 Report

Comments and Suggestions for Authors

According to the reviewers' comments, the manuscript has been carefully revised and greatly improved. Thus, I recommend the acceptance in the journal. 

Comments on the Quality of English Language

The manuscript should be carefully proofread to avoid the spelling, expression and grammar errors.